# qDSB-Seq is a general method for genome-wide quantification of DNA double-strand breaks using sequencing

Yingjie Zhu [1,8], Anna Biernacka [2,8], Benjamin Pardo[3], Norbert Dojer [1,4], Romain Forey[3], Magdalena Skrzypczak[2], Bernard Fongang[1], Jules Nde[1], Razie Yousefi[1], Philippe Pasero[3], Krzysztof Ginalski[2] & Maga Rowicka [1,5,6,7]

DNA double-strand breaks (DSBs) are among the most lethal types of DNA damage and frequently cause genome instability. Sequencing-based methods for mapping DSBs have been developed but they allow measurement only of relative frequencies of DSBs between loci, which limits our understanding of the physiological relevance of detected DSBs. Here we propose quantitative DSB sequencing (qDSB-Seq), a method providing both DSB frequencies per cell and their precise genomic coordinates. We induce spike-in DSBs by a site-specific endonuclease and use them to quantify detected DSBs (labeled, e.g., using i-BLESS). Utilizing qDSB-Seq, we determine numbers of DSBs induced by a radiomimetic drug and replication stress, and reveal two orders of magnitude differences in DSB frequencies. We also measure absolute frequencies of Top1-dependent DSBs at natural replication fork barriers. qDSB-Seq is compatible with various DSB labeling methods in different organisms and allows accurate comparisons of absolute DSB frequencies across samples.

[1] Department of Biochemistry and Molecular Biology, University of Texas Medical Branch at Galveston, 301 University Boulevard, Galveston, Texas 77555, USA. [2] Laboratory of Bioinformatics and Systems Biology, Centre of New Technologies, University of Warsaw, Zwirki i Wigury 93, 02-089 Warsaw, Poland. [3] Institut de Génétique Humaine, CNRS, Equipe Labellisée Ligue contre le Cancer, Université de Montpellier, 141 rue de la Cardonille, Montpellier 34396, France. [4] Institute of Informatics, University of Warsaw, Stefana Banacha 2, 02-097 Warsaw, Poland. [5] Institute for Translational Sciences, University of Texas Medical Branch at Galveston, 301 University Boulevard, Galveston, Texas 77555, USA. [6] Sealy Center for Molecular Medicine, University of Texas Medical Branch at Galveston, 301 University Boulevard, Galveston, Texas 77555, USA. [7] Sealy Center for Structural Biology and Molecular Biophysics, University of Texas Medical Branch at Galveston, 301 University Boulevard, Galveston, TX 77555, USA. [8] These authors contributed equally: Yingjie Zhu, Anna Biernacka. Correspondence and requests for materials should be addressed to M.R. (email: Maga.Rowicka@utmb.edu)

There is tremendous interest in precisely measuring DNA double-strand breaks (DSBs) genome-wide, as such measurement can give key insights into DNA damage and repair, cancer development[1], radiation biology, and also increasingly popular genome-editing techniques[2]. Starting with our BLESS method[3], several high-resolution and direct methods to label DSBs genome-wide have recently been developed[4–7], which have opened new possibilities for sensitive and specific DSB detection. For example, BLESS was applied in identifying the on-target and off-target cutting sites of Cas9 endonuclease[8] and studying DSB repair[9]. However, we still lack an effective strategy to both precisely detect DSB distribution genome-wide and quantify their absolute frequencies per cell, which is crucial to assess physiological relevance of detected DSBs. Immuno-fluorescence microscopy in combination with γH2A.X and 53BP1 antibodies was used to count DSBs per cell[10], but does not allow determining their precise locations. Moreover, counting discrete nuclear foci is an imprecise way to estimate numbers of DSBs per cell due to both DSB clustering and limited specificity of antibodies. Quantitative PCR (qPCR)-based methods can estimate absolute break frequency but only at selected loci[11]. An approach was developed recently to quantify breaks globally based on the amount of radiolabeled DNA and locally based on DNA break immunocapture[12], but its accuracy in detecting physiological DSBs was not tested. Another method called BLISS[7] labels DSBs by utilizing Unique Molecular Identifiers (UMIs) and thus has the potential to be quantitative. However, the total number of unique UMIs in different samples could vary highly with the sequencing depth, making UMI-based quantification inaccurate and unstable (discussed below). To solve this problem, series of libraries from the same sample were sequenced at increasing depth and mathematical modeling was used to extrapolate the true number of DSBs in the sample[7]. Such a procedure, however, adds complexity and costs to DSB sequencing and is highly dependent on the samples selected for extrapolation. Thus, despite the potential for DSB quantification, these limitations resulted in BLISS being utilized only as a DSB labeling method in projects subsequent to the first application of BLISS[13,14].

This lack of a general method and computational solution to simultaneously determine DSB frequencies per cell and their precise genomic loci limits our understanding of the physiological relevance of observed DSBs and hinders comparisons between experiments. Here we propose quantitative DSB sequencing (qDSB-Seq), an approach that allows measuring DSB frequencies per cell genome-wide and includes a computational solution to achieve accurate quantification. Our approach relies on inducing spike-in DSBs by a site-specific endonuclease, which are used to quantify DSBs detected by a DSB labeling method. Here we use i-BLESS[15] and BLESS[3], but qDSB-Seq can be combined with any DSB labeling technique. We present a comprehensive validation of qDSB-Seq in the budding yeast and show that our method gives accurate quantification results and can also be applied to human samples. We present several applications of qDSB-Seq. We quantify DSBs induced by a radiomimetic drug and characterize the resulting DSB-prone regions. We also quantify and characterize DSBs occurring during replication stress and measure absolute frequencies of Topoisomerase 1 (Top1)-dependent DSBs at natural replication fork barriers (RFBs). We reveal two orders of magnitude differences in break frequencies between the conditions we study; we also show that qDSB-Seq provides accurate comparison of absolute DSB frequencies across samples.

## Results

**qDSB-Seq implementation**. qDSB-Seq allows us to both precisely detect DSB distribution genome-wide and quantify their absolute frequencies per cell (Fig. 1a) by combining genome-wide high-resolution DSB labeling (i-BLESS[15], BLESS[3], END-seq[6], etc.) and induction of DSBs (spike-ins) in pre-determined loci using a site-specific endonuclease (Fig. 1b–e). Quantification is based on an assumption (verified below) that the number of labeled reads at a given genomic locus resulting from DSB sequencing is proportional to the underlying DSB frequency (proportionality coefficient $\alpha$ in Fig. 2a).

To estimate this coefficient $\alpha$, we induce spike-in DSBs at pre-determined genomic loci by digestion with a restriction endonuclease before DSB labeling (Fig. 1d, e). Next, relying on knowledge of exact genomic locations of spike-ins, their frequency, $B_{cut}$, is calculated from enzyme cutting efficiency, $f_{cut}$. $f_{cut}$ is calculated based on numbers of cut and uncut DNA fragments covering cutting sites in genomic DNA (gDNA) sequencing data (Fig. 1b, Fig. 2a, Methods), or from qPCR data (Supplementary Fig. 1, Methods). Finally, the absolute frequency of studied DSBs, $B_{studied}$, is estimated from DSB sequencing data:

$$B_{studied} = \frac{R_{studied}}{\alpha}, \text{ where } \alpha = \frac{R_{cut}}{B_{cut}} \qquad (1)$$

and $R_{studied}$ and $R_{cut}$ are the numbers of labeled reads originating from studied DSBs and from enzyme cutting sites (spike-ins), respectively, and $B_{cut} \sim f_{cut}$.

**Accuracy of cutting efficiency estimation**. The number of labeled reads per DSB (coefficient $\alpha$), which is used for the final DSB quantification, as explained above, is computed from enzyme cutting efficiency, $f_{cut}$ (Equation (1), Methods). Therefore, to calculate $\alpha$ accurately, we need to be able to estimate enzyme cutting efficiency precisely. Commonly, qPCR is used for precise measurement of a cutting efficiency; however, this technique is inconvenient to use for multiple cutting sites. Thus, we propose to use gDNA sequencing to determine spike-in cutting efficiencies (Fig. 2a, Methods). To verify the accuracy and reproducibility of the proposed approach, we treated immobilized and deproteinized yeast DNA with NotI enzyme and compared cutting efficiencies at its recognition sites calculated using both gDNA sequencing data and qPCR. The cutting efficiencies for the selected NotI cutting site were highly consistent: 62% for gDNA sequencing and 62% for qPCR. To examine whether our approach can also be applied to breaks introduced in vivo, which can be subjected to repair and resection, we used a yeast strain engineered to produce a single site-specific DSB by I-SceI endonuclease in vivo. Cutting efficiencies calculated based on gDNA sequencing and based on qPCR (Supplementary Fig. 1, Methods) were again very consistent: 71% and 73 %, respectively (Fig. 2b). We therefore conclude that our method of estimating enzyme cutting efficiency based on gDNA sequencing yields accurate and precise results.

**Dependence of quantification on enzyme and break type**. DSBs occurring in vivo are subject to DNA damage repair and therefore might be labeled with different efficiencies than breaks induced in vitro. Moreover, different types of double-strand DNA ends (blunt or sticky) could also be detected more or less efficiently by a given DSB labeling method. We therefore asked whether any restriction enzyme and any manner of digestion can be applied to create spike-in DSBs that would lead to accurate quantification. First, to test whether restriction enzyme choice or the type of double-strand DNA ends influences our quantification results, we determined the spontaneous DSB frequencies in yeast $G_1$-phase cells using NotI or SrfI spike-ins, which create sticky and blunt ends, respectively. The number of spontaneous breaks in $G_1$-phase cells estimated using these enzymes was consistent:

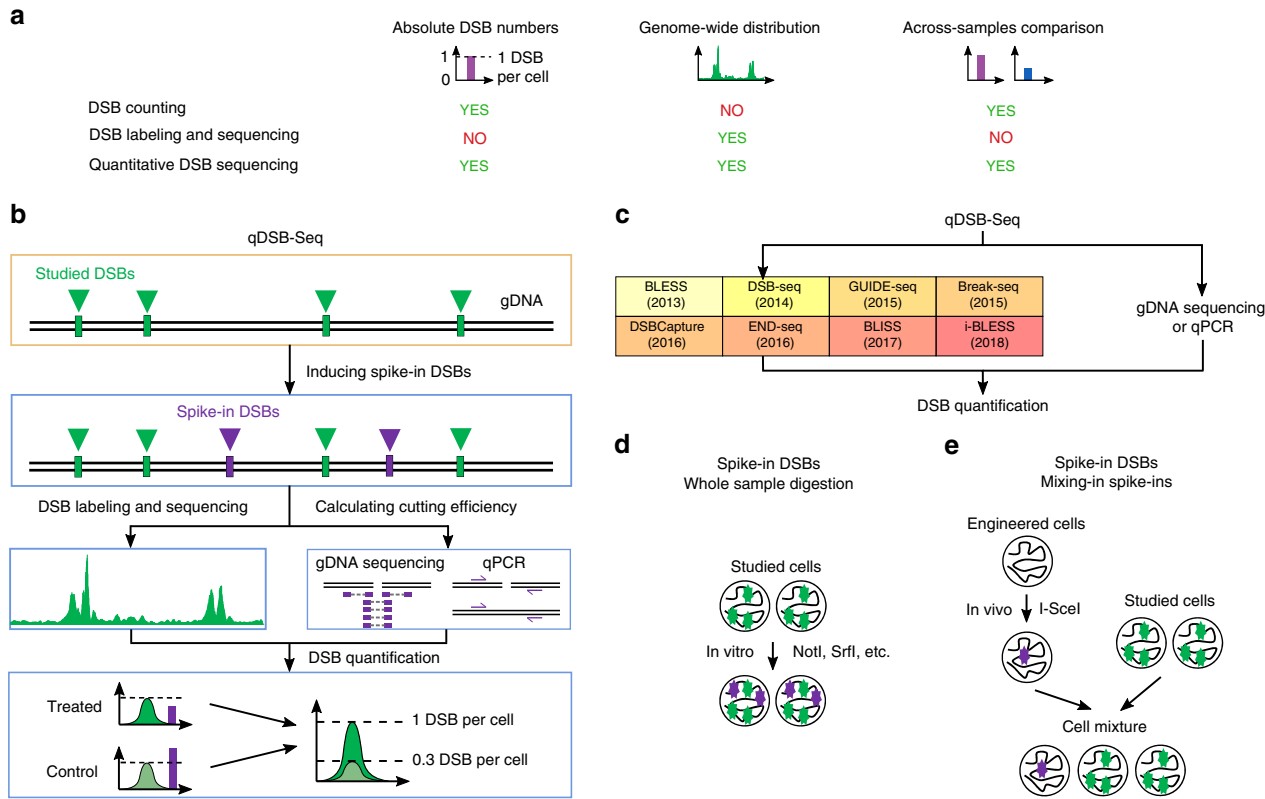

**Fig. 1** Quantitative DSB sequencing (qDSB-Seq) method. **a** A comparison of current DNA double-strand break (DSB) counting (e.g., immunofluorescence microscopy, quantitative PCR (qPCR)) and DSB sequencing strategies (e.g., BLESS[3], i-BLESS[15], END-Seq[6], Break-Seq[4], DSBCapture[5]) with our qDSB-Seq method. **b** In qDSB-Seq protocol after DSB induction cells are treated with a restriction enzyme to introduce site-specific, infrequent DSBs (spike-ins). Next, DSBs are labeled (here using i-BLESS[15] or BLESS[3]) and sequenced. Simultaneously, genomic DNA (gDNA) sequencing (or qPCR) is performed and used to estimate the cutting efficiency of the enzyme, and thus frequency of induced spike-in DSBs, which is then used to quantify the absolute DSB frequencies (per cell) of studied DSBs in the sample (Methods). **c** qDSB-Seq can be combined with any sequencing-based DSB labeling method. **d**, **e** Spike-in DSBs were induced in two different ways: **d** the studied cells were digested using the NotI, SrfI, AsiSI, or BamHI restriction enzyme in vitro; **e** cells expressing a restriction enzyme in vivo were mixed with the studied cells (I-SceI digestion) or alternatively a restriction enzyme was expressed in vivo in all studied cells (DIvA cells discussed below)

$0.9 \pm 0.3$ DSBs per cell ($n = 2$) for NotI spike-in and $1.0 \pm 0.6$ DSBs per cell ($n = 3$) for SrfI spike-in (Fig. 2c). Then, to test whether the results are affected by the manner of digestion, we compared DSB estimations based on quantification using NotI (5′ overhangs) in vitro digestion and I-SceI (3′ overhangs) in vivo digestion in hydroxyurea (HU)-treated wild-type cells (described below). Again, results were highly similar: $135 \pm 13$ (SD was estimated as described in Methods) and $153 \pm 52$ DSBs per cell ($n = 39$, Methods) (Fig. 2d). In conclusion, qDSB-Seq provided consistent results in all tested cases irrespective of the restriction enzyme used, types of DNA ends created by that enzyme, or the manner of digestion.

**Dependence of quantification accuracy on cutting efficiency.** For accurate quantification of studied DSBs, it is necessary that the relationship between the number of labeled reads and DSB frequencies at different genomic locations is linear (Equation (1), Fig. 2a). This relationship could be affected by the frequencies of spike-in DSBs, $B_{cut}$, which are determined by an enzyme cutting efficiency, $f_{cut}$. Therefore, we asked whether any frequency of induced spike-in DSBs (i.e., any enzyme -cutting efficiency) can be employed. To test the influence of enzyme cutting efficiency on the quantification results, we performed 35 digestions using enzymes with multiple cutting sites (NotI, SrfI, AsiSI, and BamHI) and then tested the linear relationship between the labeled reads and cutting efficiencies for each digestion using

Pearson's correlation coefficient. We observed that strong correlation ($R > 0.5$) (e.g., Fig. 2e) was always achieved for cutting efficiencies between 12% and 62% (Supplementary Fig. 2, Supplementary Table 1), and for some lower cutting efficiencies (4–12%). However, for the extreme cutting efficiencies (higher than 84% or lower than 4%) the correlation was always weak (Supplementary Fig. 3). In such cases, the number of observed cut or uncut fragments was low, making our estimates less accurate, which likely decreased the correlation. Moreover, small variations in $f_{cut}$ between sites contributed to the decreased correlation (Supplementary Fig. 3). Taken together, we conclude that cutting efficiencies between 12% and 62% give most accurate quantification results and no substantial bias in quantification related to break location; for frequencies between 4% and 12%, and between 62% and 84 % results vary, and using frequencies below 4% or above 84% is not recommended. We also showed above that for the optimal cutting efficiencies (12–62 %) the overall number of labeled DSB reads are highly proportional to induced DSB frequency, irrespective of genomic locations.

**Stability of estimation of DSB frequencies per cell.** We next asked whether our method generates reproducible results. To test this, we calculated DSB frequencies in untreated $G_1$-phase cells based on different spike-ins. In spite of the various enzymes used (NotI, SrfI), we obtained a very consistent number of DSBs (Fig. 2f, Supplementary Table 2). Based on our calculations the

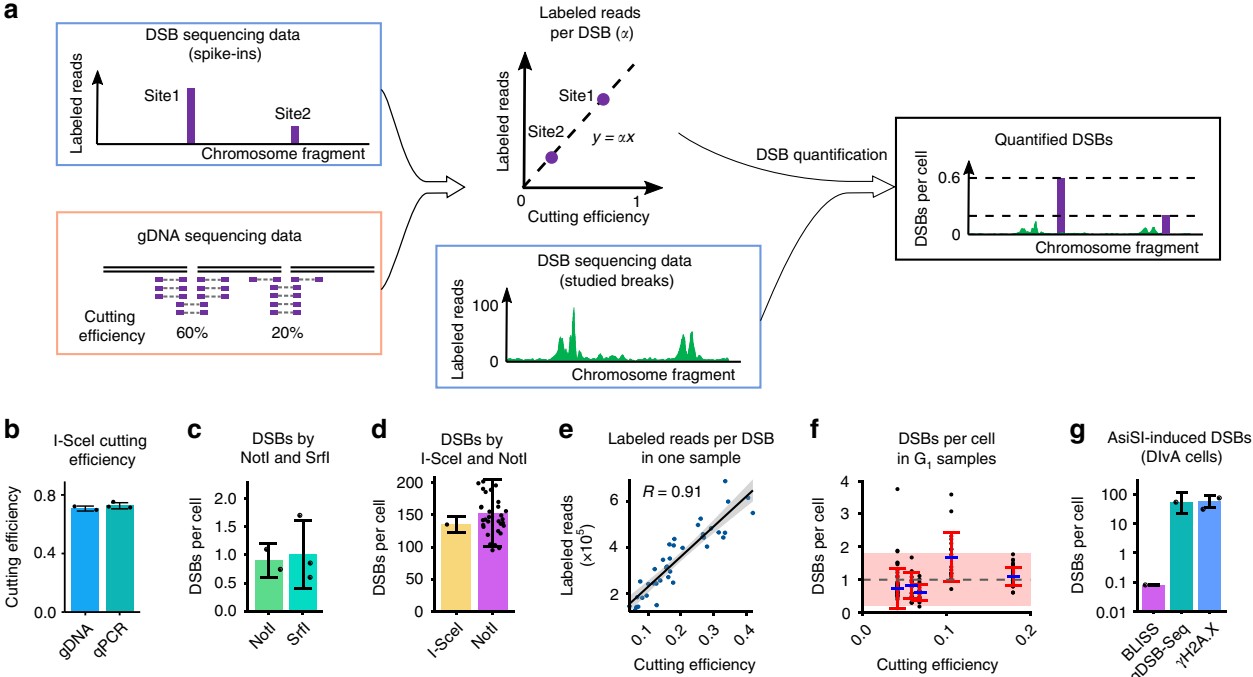

**Fig. 2** qDSB-Seq computation and validation. **a** Computation of the labeled reads per DSB and DSBs per cell. The ratio of the labeled reads and cutting efficiency at enzyme cutting sites was calculated and then used for DSB quantification in the studied genomic loci. **b** The estimation of I-SceI cutting efficiency based on gDNA sequencing ($n = 2$) and qPCR ($n = 3$). **c, d** Dependence of qDSB-Seq quantification on the restriction enzyme used. For untreated WT $G_1$-phase cells (**c**), DSBs were quantified using NotI and SrfI digestion in vitro, two and three biological replicates, respectively. For hydroxyurea (HU)-treated WT S-phase cells (**d**), DSBs were quantified by I-SceI digestion in vivo (SD was estimated as described in Methods) and NotI digestion in vitro ($n = 39$, Methods). **e** Correlation between the number of labeled reads at cutting sites and their cutting efficiencies in an untreated $G_1$-phase sample, digested with NotI enzyme with an average cutting efficiency of 18 %. $R$: Pearson's correlation coefficient. The linear regression line with 95% confidence interval is plotted. **f** DSB numbers in five untreated $G_1$-phase cells. The dashed gray line and the pink-colored area are the mean value ($n = 5$) and 95% confidence interval, respectively, for all the samples. DSBs per cell values in individual samples are denoted by blue dash, SD are denoted by red intervals ($n = 39, 17, 17, 18, 39$ from left to right, Methods). **g** Quantification of AsiSI-induced DSBs in DIvA cells by BLISS (SD, Methods), qDSB-Seq (with BLESS labeling, SD as described in Methods), and counting γH2A.X foci ($n = 2$), logarithmic scale on the $y$ axis was used. Mean and SD are shown in **b** and **c**. DSBs per cell values and SD were calculated as described in Methods in **d, f, g**. Individual data points are visualized as black dots (**b–d, f, g**). Source data are provided as a Source Data file

frequency of spontaneous DSBs in untreated $G_1$-phase wild-type cells is $1.0 \pm 0.4$ DSBs per cell ($n = 5$) (Supplementary Table 2), both the average and the range (0.6–1.7 DSBs per cell) are consistent with previous studies[16,17]. Further, we quantified DSBs based on the individual cutting sites in each of the samples. The variation of the DSB quantification results depending on the individual cutting sites used was lower than the average value (Supplementary Table 2). Similarly, in *pif1-m2* mutant, in which DSB frequency at G-quadruplex (G4) structures is increased due to lack of nuclear isoform of Pif1 DNA helicase that unwinds G4s[15], we obtained consistent average DSB number for three biological replicates ($2.1 \pm 0.3$ DSBs per cell) (Supplementary Fig. 4, Supplementary Table 2).

**Comparison of qDSB-Seq with BLISS.** Recently, a DSB labeling method called BLISS was developed, which proposes DSBs quantification by using UMIs. To assess qDSB-Seq performance relative to BLISS, we compared the abilities of both methods to quantify DSBs in DIvA (AsiSI-ER-U2OS) cells, in which DSBs were induced in vivo by activation of the restriction enzyme AsiSI upon 4-hydroxytamoxifen (4OHT) treatment[18] (Methods). To test qDSB-Seq in DIvA cells, we used BLESS for DSB labeling and analyzed resulting data to determine the interval size (±3 bp) around the AsiSI cutting sites, which was used to compute frequency of AsiSI-induced breaks based on gDNA sequencing data. To estimate AsiSI-induced DSBs, we calculated cutting efficiency

for each AsiSI cutting sites in 4OHT-treated cells and subtracted background (Methods). qDSB-Seq quantification yielded $52 \pm 65$ DSBs per cell (SD was estimated as described in Methods) consistent with $57 \pm 33$ DSBs per cell ($n = 2$) based on counting γH2A.X foci (Fig. 2g), as reported by Iannelli et al.[13] and Caron et al.[19].

To compare quantifications utilizing qDSB-Seq and BLISS[7], we used the published BLISS data[13] from DIvA cells, where DSBs were induced in the same manner as in DivA cells used for qDSB-Seq (Methods). We counted unique UMIs within ± 100 bp intervals around AsiSI cutting sites, proposed by Iannelli et al.[13] to contain reads resulting from AsiSI cutting. Next, we divided the total number of unique UMIs by the number of cells used[13] to obtain an estimate of DSBs per cell[7]. This procedure yields BLISS estimate of only 0.08 DSBs per cell, three orders of magnitude lower than $57 \pm 33$ DSBs per cell obtained from immunofluorescence[19]. In contrast, qDSB-Seq gave results consistent with immunofluorescence (Fig. 2g). This example shows that even though BLISS is a valuable tool for studying DSBs in low-input samples, application of this method for quantification is challenging and may lead to very inaccurate results. To improve quantification accuracy of BLISS, it can be combined with qDSB-Seq (Fig. 1c).

**Quantification of DSBs induced by a radiomimetic drug.** Some DSB-inducing agents affect only particular sequences and

structures, whereas others such as irradiation cause DNA damage throughout the genome. As DSB sequencing data inform only about read distribution in the genome and is primarily used to identify regions enriched in reads, even very large but global DSB induction will be undetectable using typical normalization methods, e.g., normalization to total read number or background. Therefore, to test application of qDSB-Seq to such a challenging case, we used the radiomimetic agent Zeocin[20], a member of the bleomycin drug family. After performing DSB sequencing, no apparent difference in raw read counts between Zeocin-treated (ZEO) and untreated $G_1$-phase ($G_1$) cells was observed (Fig. 3a, Supplementary Fig. 5). In contrast, after quantification (using qDSB-Seq with NotI spike-in) we concluded that $1.1 \pm 0.3$ DSBs per cell ($n = 39$, Methods) were present in the $G_1$ sample and $7.4 \pm 1.8$ in ZEO ($n = 39$, Methods), indicating that Zeocin induced $6.3 \pm 2.1$ DSBs per cell. Strikingly, Zeocin significantly increased the number of DSBs (1.7- to 13-fold) in 99.8% of 5 kb genomic intervals ($P$-value $< 2e - 12$, hypergeometric test, Methods).

Interestingly, we observed that Zeocin-induced DSBs are especially enriched (3.0-fold) in nucleosome-depleted regions (NDRs) and reduced (0.4-fold) in nucleosome-protected regions (both $P$-value $< 10^{-3}$, permutation test, Methods). Specifically, DSBs in the Zeocin-treated sample occur 1.8 times as often between predicted nucleosome positions[21] than within nucleosomes (Fig. 3b). Moreover, the preference for DSB location between

nucleosomes is even higher (4.1-fold) for long (>100 nt) NDR regions (Fig. 3c, d). However, we do not observe a 10 bp periodicity corresponding to the rotational positioning of the DNA helix on the nucleosomes. These results are consistent with previous findings that Zeocin-induced cleavage is most suppressed in nucleosome-bound DNA, and that this suppression is not dependent on inaccessibility of the minor groove, but is caused by inability of the nucleosome-bound DNA to undergo a conformational change that is required for Zeocin binding[22]. Zeocin-induced DSBs are also enriched in DNA regions capable of forming very stable DNA secondary structures (Fig. 3e), including G4s[23]. Further studies will be necessary to elucidate this phenomenon. Nevertheless, increased DNA damage on G4 structures could be related to nucleosome remodeling on G4s[24], consistent with our finding that Zeocin prefers to cleave nucleosome-free DNA.

**Quantification of DSBs induced by replication stress.** We next used qDSB-Seq to quantify replication-associated DSBs under HU-induced replication stress (Fig. 4a). HU inhibits ribonucleotide reductase, resulting in decreased dNTP levels and subsequent replication fork stalling and the slowing down of S-phase;[25] stalled forks may undergo catastrophic collapse at high concentration or prolonged HU treatment[26].

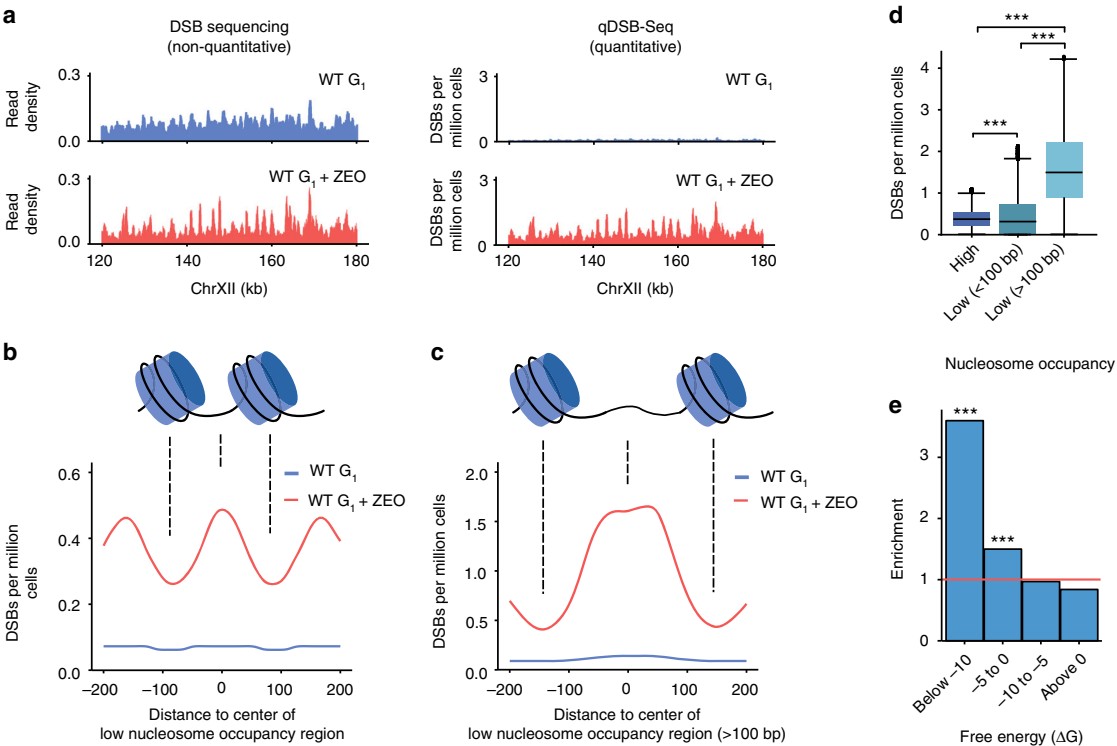

**Fig. 3** Quantification of Zeocin-induced DSBs. **a** Results of regular DSB sequencing (left, i-BLESS) and quantitative DSB sequencing (right, qDSB-Seq) in a representative fragment of chromosome XII for untreated and Zeocin-treated (ZEO) wild-type (WT) $G_1$-phase cells. For DSB sequencing, read density was normalized to the total number of reads in the sample and thus results do not allow for quantitative comparison between DSBs present in ZEO and $G_1$ samples. For qDSB-Seq, results are quantitative and DSB density within 500 bp sliding window with 50 bp step is shown. **b, c** Density of Zeocin-induced DSBs in (**b**) all and (**c**) ≥100 bp low nucleosome occupancy regions. Nucleosome locations from Lee et al.[21] were used; DSB densities, expressed as DSBs per million cells, were calculated in a 50 bp sliding window with a 5 bp step. **d** Comparison of DSB densities in high nucleosome occupancy regions (High) and low nucleosome occupancy regions (Low). Median (center line), first, and third quartiles (box limits), and 1.5 times the interquartile range (whiskers) are shown. $P$-values were calculated using two-sided Kolmogorov–Smirnov test, ***$P < 0.001$. **e** Enrichment of Zeocin-induced DSBs in regions prone to form very stable DNA secondary structures, as defined by free energy in a 50 bp sliding window as described in Methods. Zeocin-induced DSBs were defined as regions with significant enrichment of DSB-labeled reads in ZEO sample compared with $G_1$-phase control, as identified using Hygestat_BLESS. Enrichment analysis was performed using Hygestat_annotations (Methods); $P$-values were calculated using the permutation test (Methods); ***$P < 0.001$. Source data are provided as a Source Data file

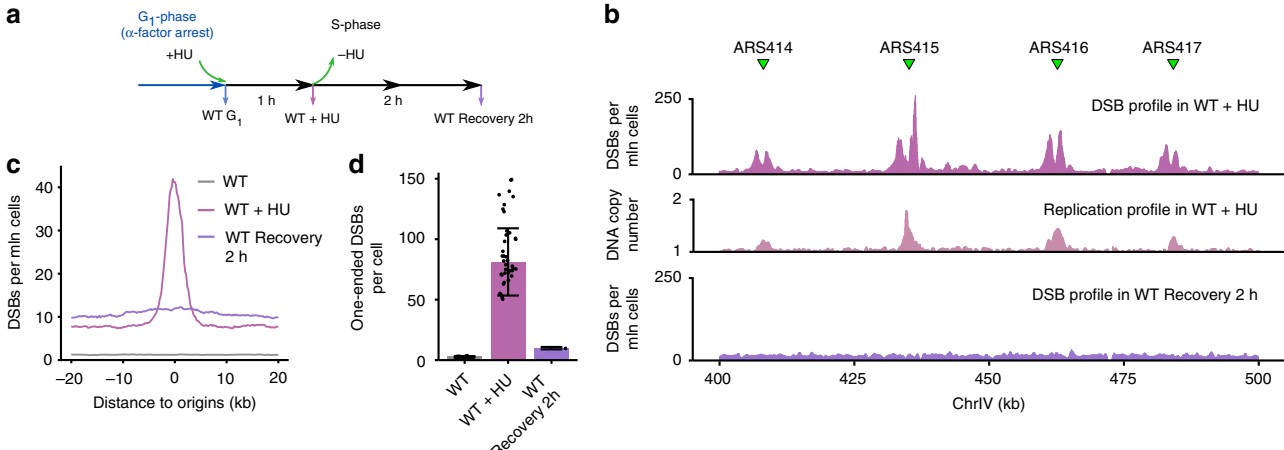

**Fig. 4** Quantification of replication stress-associated DSBs. **a** Schematic representation of hydroxyurea (HU) experiments. Cells were arrested in $G_1$-phase with α-factor, treated with HU before release to S-phase, collected after 1 h or resuspended in fresh medium and collected 2 h after removal of HU. **b** Example of quantified DSB data after 1 h HU treatment and 2 h recovery from HU. Replication origins are marked with green triangles, absolute frequencies of DSBs for a fragment of chromosome IV are shown in a million cells. As a control, replication profile in WT + HU sample (number of gDNA reads in a 500 bp in the sample normalized to untreated $G_1$-phase sample) is shown. **c** Meta-profile of DSBs around active replication origins under HU treatment, defined as 144 origins with firing time <25 min (early origins, firing time according to Yabuki et al.[39]). Median of DSB densities, expressed as DSBs per million cells in 2 kb window around each early origin, was calculated and the background was removed as described in Methods. **d** Quantification of one-ended DSBs. One-ended DSBs per cell values were calculated as described in Methods and SD for WT and WT + HU was calculated using individual NotI cutting sites (n = 39, Methods) and for Recovery 2 h using SD of background noise and assuming Poisson distribution of fragment counts (Methods). Individual data points are visualized as black dots; source data are provided as a Source Data file

Using NotI spike-in, we observed that 1 h treatment with 200 mM HU induced on average $153 \pm 52$ DSBs per cell (n = 39, Methods) in wild-type yeast cells (WT + HU sample), which represents a tenfold increase relative to untreated S-phase cells ($15 \pm 3$ DSBs per cell, n = 39, Methods). The detected breaks showed a clear replication-related pattern: a significant enrichment of DSB signal around replication origins (Fig. 4b, c). To further analyze the HU-induced DSBs, we classified them into two-ended DSBs and one-ended DSBs. One-ended DSBs are a special type of DSBs resulting from broken replication forks (Supplementary Fig. 6), whereas all other DSBs, e.g., created by endonucleases, radiation, or chemical compounds, are two-ended. We identified one-ended DSBs using our method based on comparing the number of reads between Watson and Crick strands (Supplementary Fig. 6, Methods). We discovered that among all DSBs detected in HU-treated WT cells $81.2 \pm 27.8$ DSBs (n = 39, Methods) (43%) were one-ended (Fig. 4d). Of those, 82% ($66.7 \pm 22.8$ DSBs) were located within ±10 kb regions of active replication origins, resulting in an average of 0.5 one-ended DSB per origin (Fig. 4d). The observed one-ended DSBs might correspond to broken forks resulting from transient DNA breaks occurring on the leading strand, as reported by Sasaki et al.[27]. In agreement with this theory, we discovered that 2 h after removal of HU, the number of one-ended DSBs decreased dramatically (by 88%) (Fig. 4d), indicating that replication-associated DNA damage present during HU treatment is not permanent.

**DSB quantification at ribosomal RFBs.** RFBs are natural barriers that block replication forks to protect nearby, highly expressed rRNA genes from collisions between transcription and replication complexes[27,28] (Fig. 5a). DSBs occurring at the ribosomal RFBs have been observed using Southern blotting in the budding yeast[29–32]. However, precise frequencies and genomic locations of these DSBs were not established due to lack of a quantitative and sensitive DSB detection method[27]. Using qDSB-

Seq, here we both precisely quantified DSB frequencies near RFBs and identified their genomic coordinates.

It was reported that Fob1 proteins bound to an RFB site block replication fork progression, resulting in generation of one-ended DSBs[31]. Indeed, in unperturbed S-phase cells, we observed $1.1 \pm 0.2$ DSBs per cell (n = 39, Methods; ~0.006 DSBs per rDNA repeat) on rDSB-1 and rDSB-2 sites upstream of RFB1 and RFB2 (two closely spaced RFB loci) (Fig. 5b, c and Supplementary Table 3). As expected, we did not detect any DSBs at these sites in $G_1$-arrested cells, confirming that the observed DSBs at RFBs are replication-dependent.

It was previously shown that Top1 in the presence of Fob1 specifically cleaves defined sequences in the RFB region[33]. When we inhibited the religation step of Top1 by adding 100 μM camptothecin (CPT) for 45 min, we observed a CPT-dependent DSB site (rDSB-3), exactly at the same location as the previously identified Top1-dependent cleavage site (Fig. 5c). Our quantification shows that the DSB frequency at rDSB-3 site was 0.1 DSB per cell, lower than at rDSB-1 (0.8 DSBs per cell) and rDSB-2 (0.3 DSBs per cell) (Supplementary Table 3). In addition, rDSB-3 site also colocalizes with a Fob1-binding region, in agreement with a previous discovery that the recruitment and stabilization of Top1 requires the binding of Fob1 protein[33]. Finally, our results agree with the previous work[27], which reported that approximately one DSB arises in an rDNA array during replication in a yeast cell (Fig. 5b). In conclusion, qDSB-Seq fills the need for a method enabling detection of rare breaks at RFBs and allowed us to quantify the frequency of cleavage of Top1 at RFBs.

## Discussion

We propose qDSB-Seq, a general framework that allows estimating both absolute DSB frequencies (per cell) and their precise genomic coordinates. qDSB-Seq combines a DSB-labeling method with a quantification technique; quantification is achieved by inducing easy-to-measure spike-in DSBs via restriction enzyme digestion.

Due to increasing evidence of a relationship between emergence of DSBs and human diseases such as cancer[1], there is growing interest in precise detection of DSBs. Several general genome-wide methods for detection of DSBs with single-nucleotide resolution have recently been developed[3–6]; however, their usefulness is limited because they only allow comparison of DSB levels between genomic loci within the same sample. Normalization to the total number of reads is often employed to enable comparison between different samples, but this method is not always applicable. For example, it cannot be used if DSBs are induced throughout the whole genome or if the DSB background varies, which is common[34]. Therefore, in case of agents that create such DSB patterns, e.g., irradiation or radiomimetic drugs, data normalized to the total number of reads will not reveal global induction of breaks as shown in Fig. 3a. In contrast, our approach allows not only estimation of a relative increase of DSB signal between samples (regardless of signal distribution) but also quantification of absolute DSB numbers per cell. Thus, qDSB-Seq opens up the possibility of studying, quantitatively and genome-wide, the impact of DSB inductors on genome instability, i.e., it may potentially allow determining the outcomes of different doses of anticancer drugs in healthy and tumor cells. Moreover, qDSB-Seq allows assessing DSB frequencies not only for the whole genome, but also for a specific locus. For instance, using our approach, we quantified changes of DSB frequency at RFBs between CPT-treated and -untreated cells, thus revealing the frequency of Top1-dependent DSBs in RFB region.

Several methods of DSB quantification, based on different principles, have been developed and their advantages and limitations are summarized in Table 1. BLISS[7], the recently developed DSB labeling method, allows to work with low-input samples, but it was optimized only for mammalian cells. In contrast, qDSB-Seq is very

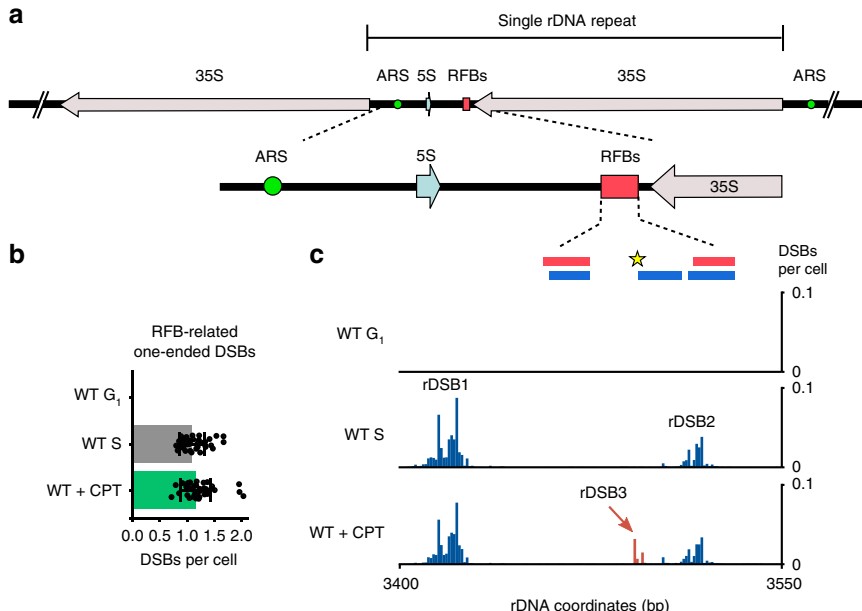

**Fig. 5** DNA double-strand breaks at replication fork barriers. **a** Scheme of replication fork barriers (RFBs) at yeast ribosomal DNA locus. **b** The total number of RFB-related one-ended DSBs (peaks as defined in **c** calculated from the difference of Watson and Crick strand reads (Methods)). DSBs per cell values and SD are shown (n = 39, Methods). Source data are provided as a Source Data file. **c** Quantified DSB signal in RFB region. RFB1 and RFB2 are indicated by the red boxes on the top. The blue boxes mark Fob1 protein binding sites mapped in vitro. The yellow star indicates Top1 cleavage site. The signal originating from rDSB-1 and rDSB-2 is presented in blue, the signal from rDSB-3 we discovered is presented in red

## Table 1 Comparison of DSB quantification methods

| Method | Description | Advantages | Limitations |
| --- | --- | --- | --- |
| qDSB-Seq | Spike-in DSBs induced at known loci used for quantification | Accuracy validated; easy to integrate with any DSB labeling method; software provided | gDNA sequencing or qPCR required; yields average DSB frequency in cell population |
| BLISS | Labels DSBs using Unique Molecular Identifiers (UMIs) | Allows low-input sample (≥1000 cells) | Proof-of-concept quantification; complex and unstable (deep sequencing of multiple libraries and modeling required); no software provided; challenging experimental technique; yields average DSB frequency in cell population |
| Immunofluorescence microscopy | Labels DSBs with antibodies, microscopy used to count nuclear foci | DSB quantification in single cells | Indirect labeling; lack of genomic coordinates; depends on antibody quality; breaks clustering hinders quantification |
| qPCR | Quantifies DSBs based on amplification of unbroken DNA fragments | Easy to perform; low cost | Local quantification (site- and sequence-specific); only works for frequent DSBs; yields average DSB frequency in cell population |
| qTUNEL | Quantifies radiolabeled dNTPs incorporated at a break site | Low cost | Cannot distinguish single-strand and double-strand DNA breaks; accuracy not validated; yields average DSB frequency in cell population |

versatile and can be used with any DSB labeling method (Fig. 1c). For example, it can be applied to yeast (in combination with i-BLESS[15] or Break-Seq[4]) or mammalian cells (in combination with, e.g., BLESS[3], END-Seq[6], Break-Seq[4], DSBCapture[5], or BLISS[7]). Moreover, to estimate the total number of DSBs in the sample using BLISS, the total number of unique UMIs in the sequencing library must be counted. Such sequencing of all labeled fragments is challenging for samples with a high number of cells or abundant DSBs and it is also expensive, as in vitro transcription and PCR amplification utilized in BLISS generate high numbers of duplicated fragments. BLISS quantification depends highly on depth of sequencing; therefore, it may give very inaccurate results, as we showed above (Fig. 2g). A proof-of-concept method to solve this problem by a mathematical modeling and deep sequencing of three libraries was proposed, but it was tested only once and not used further. In contrast, qDSB-Seq requires only partial sequencing of labeled reads, as its quantification is based on the proportion of reads originating from induced and studied DSBs. Moreover, qDSB-Seq quantification has been validated in 35 experiments and is stable and accurate (Fig. 2b–f). We also provide easy-to-use software for qDSB-Seq quantification, which works with sequencing reads from any DSB sequencing technology (Code Availability). qDSB-Seq, as a sequencing-based method, also has advantages over qPCR, which can only be used for quantification of breaks at specific loci. qPCR does not have the single-nucleotide resolution of sequencing-based methods and is only able to identify frequent DSBs. Immunofluorescence imaging, another broadly used DSB quantification technique, relies on visualizing antibodies against proteins or their specific modifications involved in the early DNA damage response, e.g., phosphorylation of the histone H2A variant, H2A.X. Immunofluorescence is an indirect method of break detection, its sensitivity and specificity depend on the quality of antibody, and it can only provide DSB numbers, but not their genomic coordinates[10]. Lastly, DSBs can be counted by the quantitative terminal deoxynucleotidyl transferase dUTP nick end labeling (qTUNEL), which quantifies radiolabeled dNTPs incorporated at the 3′-OH DNA end by the terminal deoxynucleotidyl transferase. Nevertheless, this technique cannot distinguish between single-strand breaks and DSBs, and its accuracy has not been validated[12].

Key innovation of qDSB-Seq is usage of spike-in DSBs for normalization. Such spike-in DSBs can be introduced both in vivo and in vitro; each manner of digestion has its strengths and weaknesses. In vivo digestion requires organism-specific constructs, such as the I-SceI yeast strain or DIvA (AsiSI-ER-U2OS) human cells we used[18], whereas in vitro digestion can be applied to any organism. Moreover, in case of in vivo digestion spike-in DSBs might be subjected to end resection, resulting in detection of signal in several kilobases from the original cutting site, which may complicate data analysis.

Enzyme cutting efficiency is a key parameter influencing qDSB-Seq accuracy. As shown above, usage of extremely low or high cutting efficiencies may result in inaccurate quantification results, whereas within an optimal range (12–62%) the number of labeled reads per DSB (proportionality coefficient $\alpha$) remains nearly constant, which allows for consistently accurate quantification. If spike-in DSBs are introduced in vivo, to achieve desired cutting efficiency, cells in which full digestion (or digestion with known efficiency) was performed, need to be mixed in appropriate proportions with the studied cells. In case of in vitro digestion, the studied cells should be treated with a dose of an enzyme much lower than recommended for full digestion. The enzyme cutting efficiency can be then estimated before sequencing by performing qPCR and, if needed, the experiment can be repeated with the adjusted dose.

To facilitate choice of a restriction enzyme for qDSB-Seq experiments we provide lists of restriction enzymes sorted according to their cutting efficiencies per Mb in the *Saccharomyces cerevisiae*, *Homo sapiens*, *Mus musculus*, *Caenorhabditis elegans*, *Drosophila melanogaster*, and *Arabidopsis thaliana* genomes (Supplementary Data 1), as well as Genome-wide Restriction Enzyme Digestion STatistical Analysis Tool, GREDSTAT, at http://gredstat.rowickalab.org. Enzymes with multiple cutting sites should yield best quantification results, as estimation of the enzyme cutting efficiency will be less influenced by a potential local bias. Constructs with a single enzyme cutting site, such as the I-SceI strain we employed, allow convenience of using qPCR to determine an enzyme cutting efficiency. For enzymes with multiple cutting sites, we developed a general method to estimate enzyme cutting efficiencies from gDNA sequencing data and proved its accuracy. Usage of rare cutting enzymes is preferable, as they allow for optimal cutting efficiencies at individual sites without unnecessarily increasing percentage of spike-ins in total reads. There is no benefit in using a higher spike-in percentage than necessary; high spike-in percentages, especially exceeding 30–50% of total reads, may cause quality issues with Illumina sequencing[35]. Unlike enzyme cutting efficiency, percentage of spike-in reads cannot be determined before sequencing, as it depends both on enzyme cutting efficiency and number of DSBs present in the data. Therefore, if there is a probability that high level of spike-ins may be achieved unintentionally (e.g., during pilot experiments), we recommend using our modified protocols for generation of high-quality sequencing data from low-diversity samples[35].

qDSB-Seq is compatible with any DSB labeling technique, but will also share limitations of the used method. For example, we tested that the type of generated DNA ends will not impact quantification results when using i-BLESS for DSB labeling. However, as we discussed previously[15], some DSB sequencing technologies cannot detect all types of DNA ends. Therefore, qDSB-Seq, when used in combination with such technology, will also exhibit bias in quantifying DSBs with these types of DNA ends. On the other hand, a DSB labeling technique may label not only DSBs, but structures resembling them, such as telomeres, which we take into account during data analysis (Methods).

When interpreting qDSB-Seq results, it is also important to keep in mind that qDSB-Seq relies on sequencing data derived from a population of cells. Therefore, it only yields an average number of DSBs per cell, which may or may not be representative of a typical single cell. This problem can be solved by combining qDSB-Seq with a complementary method, giving insight into population distribution of DSBs, as we proposed elsewhere[34].

In summary, qDSB-Seq allows absolute DSB quantification genome-wide and accurate cross-sample comparison, and can be applied to any organism, for which a DSB labeling method is available. qDSB-Seq relies on a key innovation, using spike-in DSBs induced by a restriction enzyme for normalization. Using qDSB-Seq, we quantified the numbers of DSBs induced by a radiomimetic drug and replication stress, measured Top1-dependent DSB frequencies at RFBs, and revealed several orders of magnitude differences in DSB frequencies. Such high variability in genome breakage highlights the importance of quantification and shows how challenging data interpretation would be without the normalization provided by qDSB-Seq. The concept of using additional information (here frequency of spike-in DSBs introduced by restriction digestion) to normalize DSB sequencing data can be also adapted to, e.g., normalizing DSB sequencing data using DSB counts based on immunofluorescence microscopy data.

## Methods

**Yeast strains and growth conditions**. Yeast strains used in this study are listed in Supplementary Table 4. Cells were grown in Yeast Extract-Peptone-Dextrose medium at 25 °C until early log phase and were then arrested in $G_1$-phase for 170 min with 8 µg mL$^{-1}$ α-factor. For exposure to Zeocin, cells were treated with 100 µg mL$^{-1}$ Zeocin (Invivogen) for 1 h. The I-SceI strain was cultured in Yeast Extract-Peptone-Raffinose medium, galactose was added for 2 h to induce I-SceI cutting. For exposure to HU, cells were released from $G_1$-phase arrest by addition of 75 µg mL$^{-1}$ Pronase (Sigma) and 200 mM HU was added 20 min before Pronase release followed by 1 h incubation. Collected cells were washed with cold SE buffer (5 M NaCl, 500 mM EDTA pH 7.5) and immediately subjected to DSB labeling.

**DIvA cell culture and induction of DSBs**. DIvA cells are U2OS cell line, created by the Gaëlle Legube group, which express the restriction enzyme AsiSI fused to the modified estrogen receptor[18]. DIvA cells were cultured in Dulbecco's modified Eagle's medium supplemented with antibiotics, 10% fetal calf serum (Invitrogen), and 1 µg mL$^{-1}$ puromycin at 37 °C under a humidified atmosphere with 5% $CO_2$[18]. The cell lines were regularly tested for mycoplasma contamination. To trigger nuclear localization of AsiSI and induce AsiSI-dependent breaks, DIvA cells were treated with 300 nM 4OHT for 4 h[18]. Untreated and 4OHT-treated DIvA cells were kindly provided by Gaëlle Legube. Collected cells were fixed with 2% formaldehyde for 30 min, washed with cold 1× phosphate-buffered saline (PBS) buffer and stored at 4 °C in 1× PBS with 0.05% $NaN_3$ until they were subjected to gDNA sequencing and DSB labeling by BLESS.

**i-BLESS labeling**. Approximately $2.5 \times 10^9$ yeast cells were resuspended in 5 mL SE buffer and mixed with 5 mL 1 % Reducta agarose (Promega) in SE buffer at 40 °C. Cell suspension was mixed with 20 mL liquid paraffin (Merck Millipore) at 40 °C and vigorously shaken by hand for 1 min, until emulsion was formed. The emulsion was then poured into 200 mL ice-cold SE buffer and the mixture was stirred for several minutes. Agarose bead suspension was gently centrifuged (200 × g, 10 min), paraffin layer was removed, and agarose bead pellet was washed three times with TE buffer. β-Mercaptoethanol (0.5 mL), 20 µL of 200 U µL$^{-1}$ lyticase solution (Sigma), and SE to a final volume of 10 mL was then added to the bead pellet, followed by 1 h incubation at 30 °C. Beads were washed with ES buffer (1% sarkosyl, 25 mM EDTA pH 8.0), resuspended in ES buffer with 50 µg mL$^{-1}$ proteinase K (Sigma), and incubated overnight at 50 °C. After incubation, the beads were washed with TE + 0.1 mM phenylmethylsulfonyl fluoride (PMSF) and twice with TE. For samples treated with restriction enzymes, the beads were washed with appropriate buffer (FastDigest buffer (Thermo Scientific) or CutSmart buffer (NEB)) followed by treatment with restriction enzymes, as described below. For samples treated with Zeocin, beads were additionally washed with NEBNext® FFPE DNA buffer and subjected to reaction with NEBNext® FFPE DNA Repair Mix for 2 h at 20 °C. Next, the beads were washed with 1 × Blunting Buffer (NEB), followed by DNA ends blunting using Quick Blunting kit (NEB) for 2 h. The beads were subsequently washed with T4 ligation buffer and then resuspended in T4 ligation buffer with 100 nM proximal adapter. After 2 h, T4 ligase was added and the beads were incubated for up to 2 days at 16 °C. After ligation, the beads were washed once with TE and encapsulated DNA was initially sonicated using Covaris S220. Total DNA was isolated using Zymoclean™ Large Fragment DNA Recovery Kit (Zymo Research) and once again fragmented by sonication to create ~400 bp fragments. Labeled fragments were captured by Dynabeads MyOne C1 beads (Invitrogen), blunted, and phosphorylated using Quick Blunting Kit (NEB), then ligated to a distal adapter (both proximal and distal adapters are identical to those used in the original BLESS method[3]). The resulting circular DNA was then linearized by I-SceI (NEB) digestion and amplified by PCR. Purified PCR products were subsequently treated with XhoI (NEB) to cleave terminal I-SceI sequences derived from adapters.

**BLESS labeling**. Fixed DIvA cells were lysed in a Lysis buffer (10 mM Tris-HCl, 10 mM NaCl, 1 mM EDTA, 1 mM EGTA, 0.2 % NP-40 pH 8), for 60 min at 4 °C and then in a Nucleus break buffer (10 mM Tris-HCl, 150 mM NaCl, 1 mM EDTA, 1 mM EGTA, 0.3% SDS pH 8), for 45 min at 37 °C. Lysed cells were resuspended in 1 × NEBuffer 2 (NEB) supplemented with 0.1% Triton X-100 and proteinase K at 100 µg mL$^{-1}$ final concentration and incubated for 2 min at 37 °C. After that, samples were immediately transferred onto ice and an equal volume of buffer supplemented with PMSF was added to quench proteinase K.

Purified nuclei were washed twice with 1 × NEBuffer 2 supplemented with 0.1% Triton X-100 and then once with blunting buffer (NEB) supplemented with 100 µg mL$^{-1}$ bovine serum albumin, followed by DNA ends blunting using Quick Blunting kit (NEB) for 45 min at room temperature. Afterwards, nuclei were washed twice with 1 × NEBuffer 2 supplemented with 0.1% Triton X-100, once with 1 × T4 ligase buffer supplemented with 0.1% Triton X-100 and once with 1 × T4 ligase buffer. Nuclei were resuspended in 1 × T4 ligase buffer with 2 µM proximal linker and in situ ligation was performed for 18–20 h at 16 °C using T4 ligase (NEB). After ligation, nuclei were washed three times with W&B buffer (5 mM Tris-HCl, 1 mM EDTA, 1 M NaCl pH 7.5, 0.1 % Triton X-100). Afterwards, gDNA was extracted by incubating nuclei in 1 × NEBuffer 2 with 0.5% Triton X-100 and proteinase K at 200 µg mL$^{-1}$ final concentration for 1 h, shaking at 65 °C, followed by isopropanol-ethanol purification. Purified gDNA was sonicated

using Covaris S220 to create ~400 bp fragments. Labeled fragments were captured by Dynabeads MyOne C1 beads (Invitrogen), blunted, and phosphorylated using Quick Blunting Kit (NEB), then ligated to a distal adapter. The resulting circular DNA was then linearized by I-SceI (NEB) digestion and amplified by PCR. Purified PCR products were subsequently treated with XhoI (NEB) to cleave terminal I-SceI sequences derived from adapters.

**Library preparation and sequencing**. Sequencing libraries for i-BLESS (yeast strains) or BLESS (DIvA cells) and respective gDNA samples were prepared using ThruPLEX DNA-seq Kit (Rubicon Genomics). i-BLESS or BLESS libraries were prepared without prior fragmentation and further size selection. Quality and quantity of the libraries were assessed on a 2100 Bioanalyzer using HS DNA Kit (Agilent) and on a Qubit 2.0 Fluorometer using Qubit dsDNA HS Assay Kit (Life Technologies). The libraries were sequenced (at least 2 × 70 bp) on Illumina HiSeq2000/2500/4000 platforms, according to our modified experimental and software protocols for generation of high-quality data from low-diversity samples[35].

**qDSB-Seq with in vitro digestion**. In addition to DSB sequencing, as described above, a digestion with a restriction enzyme was performed after proteinase K treatment and before DSB labeling. Samples were treated with NotI (NEB, Thermo Scientific), SrfI (NEB), AsiSI (NEB), or BamHI (Thermo Scientific) for 1 h at 37 °C.

**qDSB-Seq with I-SceI spike-in**. For I-SceI spike-in we used a yeast strain (I-SceI strain) with GAL-inducible I-SceI endonuclease and a single I-SceI-cutting site integrated at the ADH4 locus on chromosome VII. To measure the cleavage efficiency of I-SceI, cell aliquots were taken before (RAFF) and 2 h after (GAL) cleavage induction, and total gDNA was extracted. DNA was serially diluted and amplified for 25 cycles with primers spanning the I-SceI cutting site. Cleavage efficiency was inferred by comparing the amount of amplified DNA in GAL (cut) vs. RAFF (uncut) conditions. We used CASY Cell Counter (Roche Applied Science) to mix this spike-in with our sample of interest (wild-type cells with replication stress induced by HU treatment) in proportion 2:98. The cutting ratio of the I-SceI endonuclease expressed in the I-SceI strain was estimated using an unmixed I-SceI strain and Equation (2) below.

**Quantitative PCR**. To validate cutting efficiency for NotI, input gDNA was analyzed by real-time PCR using primers flanking a selected NotI site at chrI: 114016–114023 (forward: 5′-AGAGTTGGGAATGTGTGCCC-3′, reverse: 5′-GGG CAGCAACACAAAGTGTC-3′) and KAPA SYBR® FAST kit (Life Technologies). Four technical replicates using two different concentrations of input DNA were performed. We compared the amount of PCR product amplified in untreated (U) vs. NotI-treated cells (N) by data analysis based on the $\Delta C_T$ method[36], where the $\Delta C_T$ was obtained by subtraction of the threshold cycle $C_T$ in sample U from the $C_T$ in sample N: $\Delta C_T = C_T (N) - C_T (U)$. Final cutting efficiency was calculated as mean efficiency for all dilutions according to the formula below:

$$f_{cut} = 1 - \frac{1}{2^{\Delta C_T}}. \qquad (1)$$

We used calibration data to empirically correct $\Delta C_T$.

**Sequencing data analysis**. We used iSeq (http://breakome.eu/software.html) to ensure sequencing data quality before mapping. Next, iSeq was used to remove i-BLESS or BLESS proximal and distal barcodes (5′-TCGAGGTAGTA-3′ and 5′-TCGAGACGACG-3′, respectively). Reads labeled with the proximal barcode, which are directly adjacent to DSBs, were selected and mapped to the version of the yeast S288C genome sacCer3 (we manually corrected common polymorphisms) or to the human genome GRCh37 using bowtie[37] v0.12.2 with the alignment parameters "-m1 –v1" (to exclude ambiguous mapping and low-quality reads). For ribosomal DNA mapping in RFB analysis, we mapped sequencing reads using the parameter "-v1" to allow multiple mapped reads. The end base pairs of the reads were trimmed using bowtie "−3" parameter. The parameter choice was based on the iSeq quality report. For calculation of the absolute number of DSBs per cell only mapped reads were retained. Further, the reads identified as originating from telomere ends were removed. For the yeast data, the telomeric reads were identified as those exhibiting the CAC motif in the whole AC-rich strand; regular expression C{0,3}AC{1,10} in the PERL language was used to identify them.

**Calculation of DSB frequencies per cell**. Paired-end sequencing of gDNA or qPCR was used to measure the cutting efficiency of an endonuclease. For an enzyme with a single cutting site (e.g., I-SceI), we used the following procedure to calculate cutting efficiency ($f_{cut}$) from whole genome paired-end sequencing data:

$$f_{cut} = \frac{N_{cut}}{N_{cut} + 2N_{uncut}} - f_{bg}, \qquad (2)$$

where $N_{cut}$ is the number of fragments cut by an enzyme, $N_{uncut}$ is the number of uncut fragments covering the cutting site, and $f_{bg}$ is the background level of breaks (e.g., resulting from sonication). $N_{cut}$ fragments were counted in empirically determined, several nucleotide vicinities of the canonical cutting sites, based on

visual examination of the read distribution. For enzymes with multiple cutting sites, reads mapped to each cutting site were first classified as cut or uncut and the results were summed over all cutting sites ($N_{sites}$):

$$f_{cut} = \frac{\sum_{i=1}^{N_{sites}} N_{cut}^i}{\sum_{i=1}^{N_{sites}} N_{cut}^i + 2\sum_{i=1}^{N_{sites}} N_{uncut}^i} - f_{bg} \qquad (3)$$

To estimate cutting efficiency, we used only cutting sites to which >100 paired-end reads (in yeast) were mapped and their cutting efficiency was larger than 0. To estimate background break frequency, $f_{bg}$, we randomly selected 1000–2000 genomic windows of the same size as those used to count cut and uncut fragments and estimated cutting efficiency in those intervals using the formula, $f_{bg} = \frac{N_{cut}}{N_{cut} + 2N_{uncut}}$. For clarity, these errors are omitted in Equations (4) to (6).

Next, we calculated the number of spike-in DSBs induced at restriction sites, $B_{cut}$:

$$B_{cut} = f_{cut} N_{sites} p, \qquad (4)$$

where $f_{cut}$ is the cutting efficiency in undiluted samples, $N_{sites}$ is the number of used enzyme restriction sites (e.g., 39 for NotI), and $p$ is the proportion of digested cells ($p = 1$ unless mixing with an in vivo digested construct is used).

Then we computed the number of mapped sequencing reads per DSB or the coefficient, $\alpha$:

$$\alpha = \frac{R_{cut}}{B_{cut}}, \qquad (5)$$

where $R_{cut}$ is the number of labeled reads mapped to the cutting sites.

Finally, we computed studied DSBs per cell ($B_{studied}$) using the following formula:

$$B_{studied} = \frac{R_{studied}}{\alpha} \qquad (6)$$

where $B_{studied}$ is the number of studied DSBs per cell in the whole genome or in a specific region (e.g., a replication region), or at a specific location (e.g., an enzyme cutting site). In this study, we calculated the studied breaks per cell for the whole genome after subtracting reads generated from enzyme cutting sites, telomeres, and ribosomal DNA. Error of $B_{studied}$ is the SD of breaks calculated from different cutting sites for enzymes with multiple cutting sites (Supplementary Table 2). For example, for NotI, which has 39 cutting sites in the yeast strain we used, SD was calculated using results of DSB quantifications based on individual NotI cutting sites. By comparing with SD calculated from all cutting sites in different replicates, we concluded that SD calculated based on individual cutting sites is a conservative estimate of SD of $B_{studied}$ (Supplementary Table 2). For an enzyme with a single cutting site in a given genome (e.g., I-SceI strain) or when the signal from individual enzyme cutting sites was weak (DIvA cells), we estimated SD of $f_{cut}$ and used it to calculate SD of $B_{studied}$. Namely, we estimated SD of $f_{cut}$ using the formula $\sqrt{\sigma_{bg}^2 + \sigma_{Poisson}^2}$, where $\sigma_{bg}$ is SD of $f_{bg}$, calculated as above, and $\sigma_{Poisson}$ is calculated assuming Poisson distribution of cut and uncut fragment counts ($N_{cut}$ and $N_{uncut}$) and is approximated using the formula:

$$\frac{1}{2}\left( \frac{N_{cut} + \sqrt{N_{cut}}}{(N_{cut} + \sqrt{N_{cut}}) + 2(N_{uncut} - \sqrt{N_{uncut}})} - \frac{N_{cut} - \sqrt{N_{cut}}}{(N_{cut} - \sqrt{N_{cut}}) + 2(N_{uncut} + \sqrt{N_{uncut}})} \right). \qquad (7)$$

**Background estimation and removal**. To quantify DSBs likely resulting from broken forks near origins, we first removed background not related to replication. To define such background, we calculated DSB density in a 500 bp sliding window with a 50 bp step, the peak of this distribution was assumed to be background DSB frequency. This background was subtracted from the data at each position, resulting negative values were assigned to zero.

**Calculation of DSBs per cell in BLISS data**. We downloaded BLISS data (NCBI SRA accession SRR5441980[13]) from DIvA cells treated by 4OHT in the same manner as our gDNA and BLESS samples. We scanned the R1 reads in the BLISS data for 8 bp UMIs and the sample barcode 5′-CATCACGC-3′ at the beginning of the reads (i.e., for reads originating directly from DSBs) and retained the reads with both the UMI and the barcode. We trimmed the 16 bp prefix sequence from the R1 reads, the trimmed sequences were mapped to the human genome GRCh37 using bowtie[37] v0.12.2 with the alignment parameters "-m1 –v1." Next, using in-house PERL script, we counted the read depth of unique UMIs for each genomic position using the first nucleotide of each mapped reads (only one sequence with same UMIs at a genomic position was retained). Finally, the number of unique UMIs around AsiSI cutting sites was counted in a ±100 bp interval to calculate DSBs induced by AsiSI, as proposed by Iannelli et al.[13]. Reads from AsiSI sites closer than 200 bp were additionally processed to avoid double-counting, reads mapping to chrY sites were rejected since U2OS cells originate from a female. To calculate AsiSI-induced DSBs per cell, we divided the total number of unique UMIs originating from 1202 AsiSI sites by the reported number of cells used by Iannelli et al.[13]. SD for results of BLISS quantification was estimated assuming the Poisson distribution of UMI and cell

counts and using the formula $\frac{1}{2}\left( \frac{N_{UMI} + \sqrt{N_{UMI}}}{N_{cell} - \sqrt{N_{cell}}} - \frac{N_{UMI} - \sqrt{N_{UMI}}}{N_{cell} + \sqrt{N_{cell}}} \right)$, where $N_{UMI}$ is the number of UMI reads and $N_{cell}$ is the number of cells used.

**Analysis of fragile regions and enrichment**. Hygestat_BLESS v1.2.3 in the iSeq package was used to identify fragile regions (i.e., regions with significant increase of the read numbers in treatment versus control samples), which were defined using the hypergeometric probability distribution and Benjamini–Hochberg correction. To evaluate the enrichment of fragile regions on nucleosomes, we used Hygesta-t_annotations v2.0, which computed the proportion of mappable nucleotides belonging to both the fragile regions and the nucleosomes, and the proportion of mappable nucleotides belonging to both genomic regions and the nucleosomes. To estimate the P-value for the feature enrichment inside fragile regions, we used 1000 permutations to calculate the empirical distribution of the ratio under the null hypothesis.

**Identification of one-ended DSBs**. To estimate the total number of one-ended DSBs, we performed hypergeometric test based on the number of i-BLESS sequencing reads from Watson and Crick strands using Hygestat_BLESS v1.2.3 in the iSeq package with a 500 nt window size. Regions with $P < 1e − 10$ for enrichment of either Watson or Crick strand reads were classified as one-ended DSB regions. The Bonferroni correction was used to compute P-values. The difference between numbers of reads from Watson and Crick was used to calculate the number of one-ended DSBs using the DSB quantification method described above.

**Comparison of DSB levels between ZEO and $G_1$ samples**. We used read counts for 5000 nt mappable intervals produced by Hygestat_BLESS; ZEO read numbers were normalized using qDSB-Seq quantification. We evaluated the null hypothesis that the number of DSBs in $G_1$-phase cells is the same or lower than in ZEO using very conservative 5 SD confidence intervals (assuming Poisson distribution of reads). All genomic windows with >17 reads in 5 kb were significantly enriched in DSBs in ZEO as compared with $G_1$-phase cells ($P < 2e − 12$, calculated using the hypergeometric probability distribution and the Bonferroni correction).

**DNA secondary structure prediction**. DNA secondary structures were defined by free energy at 37 °C using UNAFold[38] v3.8 in a 50 bp sliding window with a 25 bp step along the whole yeast genome.

## Statistical analysis

Results of quantification are shown as mean ± SD or DSBs per cell and their SD values as described in Methods. To conduct enrichment analysis, the P-values were first calculated using the hypergeometric distribution function as implemented in the GNU Scientific Library for C++ and then corrected for multiple hypothesis testing using the Benjamini–Hochberg method. The threshold for statistical significance was $P < 0.05$.

## Reporting summary

Further information on research design is available in the Nature Research Reporting Summary linked to this article.

## Data availability

The datasets generated by DSB sequencing and gDNA sequencing were submitted to NCBI Sequence Read Archive (SRA) under the accession numbers SRP189465 and SRP125409. The BLISS data used was obtained from NCBI SRA under the accession number SRR5441980. The source data underlying Figs. 2b–g, 3d, e, 4d, and 5b, Supplementary Figs. 1b-c, 2, and 4, and Supplementary Table 2 are provided as a Source Data file. All other data are available from the authors upon reasonable request.

## Code availability

qDSB-Seq software is available at https://github.com/rowickalab/qDSB-Seq. iSeq software used in this study is available upon request from authors or at http://breakome.eu/software.html.

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

## Acknowledgements

This research was supported by the NIH grant R01GM112131 to M.R. (Y.Z., N.D., B.F., J. N., R.Y., and M.R.), Polish National Science Centre grant 2015/17/D/NZ2/03711 to M.S., and Foundation for Polish Science grant TEAM/2016–2 to K.G. This work was also supported by Ligue contre le Cancer (Equipe labelisee), Agence Nationale pour la Recherche (ANR), and Institut National du Cancer (INCa) grants to P.P. (B.P., R.F., and P.P.), National Science Center grant 2016/21/B/ST6/01471 to N.D., and a training fellowship from the Gulf Coast Consortia on the Computational Cancer Biology Training Program (CPRIT Grant Number RP170593) to Y.Z. We are grateful to Gaëlle Legube for kindly providing DIvA cells and for helpful discussions, to Andrzej Kudlicki for helpful discussions, and to Heather Lander of the Sealy Center for Structural Biology and Molecular Biophysics at UTMB, for editorial services for the manuscript.

## Author contributions

M.R. conceived qDSB-Seq and supervised and coordinated the project. K.G. supervised i-BLESS, BLESS, and NGS experiments. Y.Z., A.B., and M.R. wrote the manuscript. B.P., M.S., P.P., and K.G. edited the manuscript. Y.Z. performed data analysis and developed software. N.D. performed initial data analysis and developed software. A.B., B.P., M.S., P.P., K.G., and M.R. designed experiments. A.B. and M.S. performed i-BLESS and qDSB-Seq experiments. A.B., B.P., and R.F. prepared cells. Y.Z. prepared figures. B.F., J.N., and R.Y. contributed to software development and data analysis. M.S. designed and performed qPCR, library preparation, and next-generation sequencing. All authors read the manuscript.

## Additional information

**Competing interests:** The authors declare no competing interests.

