## [Peer Review File · Nature Communications]

Reviewers' Comments:

Reviewer #1:

Remarks to the Author:

The genome-wide detection of DNA double strand breaks (DSBs) genome-wide is a very exciting area of research that recently became especially important due to increasing evidence of a relationship between emergence of DSBs and human diseases including cancer. Several genome-wide methods for detection of DSBs have been previously developed, but their usefulness is limited because they only allow comparison of DSB levels between genomic loci within the same sample. The study of Zhu et al introduces a new quantitative DSB sequencing (qDSB-Seq) method allowing determining both DSB frequencies per cell and their precise genomic locations. The key innovation of the qDSB-Seq is usage of spike-in DSBs for normalization. In particular, the authors introduce spike-in DSBs by a site-specific endonuclease (in vivo or in-vitro) and use them to quantify detected DSBs (labelled e.g. using –BLESS). The authors utilize their qDSB-Seq method to determine the number of DSBs induced by radiomimetic drug and replication stress and demonstrate two orders of magnitude differences in DSB frequencies. The authors also measured for the first time absolute frequencies of Top1-dependent DSBs at natural replication fork barriers. Overall, the study by Zhu et al will be of great interest for the diverse readership of "Nature communications" journal because it provides researchers with a new exciting tool and also because using this tool the authors present their exiting new data characterizing the amount and precise locations of DSBs genome-wide. Importantly, the qDSB-Seq method is fully validated by authors by several different methods and a variety of conditions are described (e.g. in-vivo and in-vitro spike-in DSBs and their comparison to each other). In terms of new exciting findings, of great importance is the quantification of zeocin-induced DSBs and the demonstration of the dependence of zeocin-induced breakage on the nucleosome occupancy, quantification of replication-associated DSBs, and characterization of the frequencies of DSBs formed at yeast ribosomal DNA locus. Together, the study performed by Dr. Maga Rowicka's group represents an exciting and very timely investigation and will be of great interest for the researchers investigating the mechanisms of DNA repair, genomic instability, genetic recombination, cancer etiology and the maintenance of genomic integrity.

Reviewer #2:

Remarks to the Author:

The authors previously introduced BLESS for identifying sites of double-strand breaks genome-wide, followed by various improvements in subsequent publications. Here they show that a "spike-in" strategy consisting of introducing low-frequency cleavages using various restriction enzymes allows for calibration of BLESS data, allowing them to calculate the absolute frequency of ds breaks per cell. The effectiveness of the method is supported by application of the method to a variety of treatments that introduce double-strand breaks. Technically, the paper is solid, and the approach is generally applicable to determinations of absolute ds break frequency. However, whether or not it will be used depends in part on whether BLESS is preferred over competing methods, for example BLISS, which has the advantage of being applicable to much lower cell numbers, or optical counting methods. I would be more enthusiastic about this work if the authors had provided evidence that BLESS with their calibration strategy can outperform BLISS, which was shown to also provide ds breaks per cell measurements, or at least explain why their method would be preferred over the various competing methods. Otherwise, potential users of the method will not have sufficient basis to decide on one method over another, and so publication in Nature Communications as opposed to a specialized journal is not recommended.

Reviewers' comments:

Reviewer #1 (Remarks to the Author):

The genome-wide detection of DNA double strand breaks (DSBs) genome-wide is a very exciting area of research that recently became especially important due to increasing evidence of a relationship between emergence of DSBs and human diseases including cancer. Several genome-wide methods for detection of DSBs have been previously developed, but their usefulness is limited because they only allow comparison of DSB levels between genomic loci within the same sample. The study of Zhu et al introduces a new quantitative DSB sequencing (qDSB-Seq) method allowing determining both DSB frequencies per cell and their precise genomic locations. The key innovation of the qDSB-Seq is usage of spike-in DSBs for normalization. In particular, the authors introduce spike-in DSBs by a site-specific endonuclease (in vivo or in-vitro) and use them to quantify detected DSBs (labelled e.g. using –BLESS). The authors utilize their qDSB-Seq method to determine the number of DSBs induced by radiomimetic drug and replication stress and demonstrate two orders of magnitude differences in DSB frequencies. The authors also measured for the first time absolute frequencies of Top1-dependent DSBs at natural replication fork barriers. Overall, the study by Zhu et al will be of great interest for the diverse readership of “Nature communications” journal because it provides researchers with a new exciting tool and also because using this tool the authors present their exiting new data characterizing the amount and precise locations of DSBs genome-wide. Importantly, the qDSB-Seq method is fully validated by authors by several different methods and a variety of conditions are described (e.g. in-vivo and in-vitro spike-in DSBs and their comparison to each other). In terms of new exciting findings, of great importance is the quantification of zeocin-induced DSBs and the demonstration of the dependence of zeocin-induced breakage on the nucleosome occupancy, quantification of replication-associated DSBs, and characterization of the frequencies of DSBs formed at yeast ribosomal DNA locus. Together, the study performed by Dr. Maga Rowicka’s group represents an exciting and very timely investigation and will be of great interest for the researchers investigating the mechanisms of DNA repair, genomic instability, genetic recombination, cancer etiology and the maintenance of genomic integrity.

Response: Thank you for taking time to review our paper and for your remarks. We are glad that you appreciate the quality of our work and exciting tools and results we want to share with Nature Communications readership.

Reviewer #2 (Remarks to the Author):

The authors previously introduced BLESS for identifying sites of double-strand breaks genome-wide, followed by various improvements in subsequent publications. Here they show that a "spike-in" strategy consisting of introducing low-frequency cleavages using various restriction enzymes allows for calibration of BLESS data, allowing them to calculate the absolute frequency of ds breaks per cell.

Response: Thank you for taking time to review our paper and for your remarks. qDSB-Seq is not only a strategy for quantifying BLESS and i-BLESS data, it can be

as well used with any sequencing-based DSB labeling method (i.e. END-Seq, DSB-Capture, Break-Seq, GUIDE-Seq, BLISS), which is an important advantage of our approach. To make this point clearer, we added panel **1c** to **Figure 1**.

The effectiveness of the method is supported by application of the method to a variety of treatments that introduce double-strand breaks. Technically, the paper is solid, and the approach is generally applicable to determinations of absolute ds break frequency. However, whether or not it will be used depends in part on whether BLESS is preferred over competing methods, for example BLISS, which has the advantage of being applicable to much lower cell numbers, or optical counting methods. I would be more enthusiastic about this work if the authors had provided evidence that BLESS with their calibration strategy can outperform BLISS, which was shown to also provide ds breaks per cell measurements, or at least explain why their method would be preferred over the various competing methods. Otherwise, potential users of the method will not have sufficient basis to decide on one method over another, and so publication in Nature Communications as opposed to a specialized journal is not recommended.

Response: We added comprehensive discussion (below and pages 16-18) of advantages and limitations of various DSB quantification strategies (including BLISS and optical counting methods) and summarized them in Table 1, which clearly shows superiority of qDSB-Seq over other techniques. Specifically, while BLISS is a valuable tool for studying DSBs in low-input samples, application of this method for quantification is challenging and complicated. In particular, it requires sequencing series of libraries from the same sample at increasing depth and using mathematical modeling to extrapolate the true number of DSBs in the sample. Such a procedure is complex, expensive and highly dependent on the sequencing libraries selected for extrapolation. It should be also noted that BLISS was used for quantification only once in the paper reporting the method; in the two subsequent papers, co-authored by BLISS creators, BLISS was used only for DSB labeling. In our opinion, it indicates that BLISS is not likely to become the method of choice for DSB quantification.

In contrast, qDSB-Seq has been comprehensively validated and proved accurate, stable, and robust. qDSB-Seq is compatible with any DSB labeling methods, which makes it universal, i.e. it can be combined with BLISS, enabling work with low number of cells. In addition, we provide detailed computational method and software which makes DSB quantification easy (<https://github.com/rowickalab/qDSB-Seq>).

Moreover, below and on page 9 in the manuscript, we provide direct comparison of qDSB-Seq with BLISS. Both methods are used for quantification of AsiSI-induced DSBs in DivA cells. This comparison shows that qDSB-Seq clearly outperforms BLISS: qDSB-Seq quantification is consistent with immunofluorescence results while BLISS quantification is three orders of magnitude off (**Figure 2g**), proving that quantification using BLISS may lead to very inaccurate results.

“Comparison of qDSB-Seq with BLISS. Recently a new DSB-labelling method called BLISS was developed, which proposes DSBs quantification by using Unique Molecular Identifiers (UMIs). To assess qDSB-Seq performance relative to BLISS, we compared the abilities of both methods to quantify DSBs in DivA (AsiSI-ER-U2OS) cells, in which DSBs were induced *in vivo* by activation of the restriction enzyme

AsiSI upon 4-hydroxytamoxifen (4OHT) treatment¹⁸ (**Methods**). To test qDSB-Seq in DivA cells, we used BLESS for DSB labeling and analyzed resulting data to determine the interval size (± 3 bp) around the AsiSI cutting sites, which was used to compute frequency of AsiSI-induced breaks based on gDNA sequencing data. To estimate AsiSI-induced DSBs, we calculated cutting efficiency for each AsiSI cutting sites in 4OHT-treated cells and subtracted background (**Methods**). qDSB-Seq quantification yielded 52 ± 65 DSBs per cell consistent with 57 ± 33 DSBs per cell based on counting γ H2A.X foci (**Fig. 2g**), as reported by Iannelli *et al.*¹³ and Caron *et al.*¹⁹

To compare quantifications utilizing qDSB-Seq and BLISS⁷, we used the published BLISS data¹³ from DivA cells, where DSBs were induced in the same manner as in DivA cells used for qDSB-Seq (**Methods**). We counted unique UMIs within ± 100 bp intervals around AsiSI cutting sites, proposed by Iannelli *et al.*¹³ to contain reads resulting from AsiSI cutting. Next, we divided the total number of unique UMIs by the number of cells used¹³ to obtain an estimate of DSBs per cell⁷. This procedure yields BLISS estimate of only 0.08 DSBs per cell, three orders of magnitude lower than 57 ± 33 DSBs per cell obtained from immunofluorescence¹⁹. In contrast, qDSB-Seq gave results consistent with immunofluorescence (**Fig. 2g**). This example shows that even though BLISS is a valuable tool for studying DSBs in low-input samples, application of this method for quantification is challenging and may lead to very inaccurate results. To improve quantification accuracy of BLISS, it can be combined with qDSB-Seq (**Fig. 1 c**).”

“Several methods of DSB quantification, based on different principles, have been developed and their advantages and limitations are summarized in **Table 1**. BLISS⁷, the recently developed DSB labelling method, allows to work with low-input samples, but it was optimized only for mammalian cells. In contrast, qDSB-Seq is very versatile and can be used with any DSB labeling method (**Fig. 1c**). For example, it can be applied to yeast (in combination with i-BLESS¹⁵ or Break-Seq⁴) or mammalian cells (in combination with e.g. BLESS³, END-Seq⁶, Break-Seq⁴, DSB-Capture⁵ or BLISS⁷). Moreover, to estimate the total number of DSBs in the sample using BLISS, the total number of unique UMIs in the sequencing library must be counted. Such sequencing of all labeled fragments is challenging for samples with a high number of cells or abundant DSBs, and it is also expensive since *in vitro* transcription and PCR amplification utilized in BLISS generate high numbers of duplicated fragments. BLISS quantification depends highly on depth of sequencing, therefore it may give very inaccurate results, as we showed above (**Fig. 2g**). A proof-of-concept method to solve this problem by a mathematical modeling and deep sequencing of three libraries was proposed, but it was tested only once and not used further. In contrast, qDSB-Seq requires only partial sequencing of labeled reads since its quantification is based on the proportion of reads originating from induced and studied DSBs. Moreover, qDSB-Seq quantification has been validated in 35 experiments and is stable and accurate (**Fig. 2 b-f**). We also provide easy-to-use software for qDSB-Seq quantification, which works with sequencing reads from any DSB sequencing technology (**Code Availability**). qDSB-Seq, as a sequencing-based method, also has advantages over qPCR, which can only be used for quantification of breaks at specific loci. qPCR does not have the single-nucleotide resolution of sequencing-based methods and is only able to identify frequent DSBs. Immunofluorescence imaging, another broadly used DSB quantification technique, relies on visualizing

antibodies against proteins or their specific modifications involved in the early DNA damage response, e.g. phosphorylation of the histone H2A variant, H2A.X. Immunofluorescence is an indirect method of break detection, its sensitivity and specificity depend on the quality of antibody and it can only provide DSB numbers, but not their genomic coordinates¹⁰. Lastly, DSBs can be counted by qTUNEL, which quantifies radiolabeled dNTPs incorporated at the 3'-OH DNA end by the terminal deoxynucleotidyl transferase. Nevertheless, this technique cannot distinguish between single-strand and double-strand breaks and its accuracy has not been validated¹²."

Table 1. Comparison of DSB quantification methods.

Method	Assay characteristics	Advantages	Limitations
qDSB-Seq	Spike-in DSBs induced at known loci used for quantification	Accuracy validated; easy to integrate into any DSB labeling method; software provided	gDNA sequencing or qPCR required; yields average DSB frequency in cell population
BLISS	Labels DSBs using Unique Molecular Identifiers (UMI)	Allows low-input sample (≥ 1000 cells)	Proof-of-concept quantification; complex and unstable (deep sequencing of multiple libraries and modeling required); no software provided; challenging experimental technique; yields average DSB frequency in cell population
Immunofluorescence microscopy	Labels DSBs with antibodies, microscopy used to count nuclear foci	DSB quantification in single cells	Indirect labeling; lack of genomic coordinates; depends on antibody quality; breaks clustering hinders quantification;
qPCR	Quantifies DSBs based on amplification of unbroken DNA fragments	Easy to perform; Low cost	Local quantification (site- and sequence-specific); only works for frequent DSBs; yields average DSB frequency in cell population
qTUNEL	Quantifies radiolabeled dNTPs incorporated at a break site	Low cost	Cannot distinguish single-strand and double-strand DNA breaks; accuracy not validated; yields average DSB frequency in cell population

Reviewers' Comments:

Reviewer #2:

Remarks to the Author:

The authors have addressed my concerns well.